# 'Who wants to be a teacher in Ghana?' A structural equation modelling analysis of prospective language teachers' behavioural intentions to pursue a career in teaching

Ernest Nyamekye[1‡*], John Zengulaaru[2‡]

**1** Department of Arts Education, University of Cape Coast, Cape Coast, Ghana, **2** Department of Social Studies Education, University of Education Winneba, Winneba, Ghana

‡ These authors are joint senior authors on this work.
* ernest.nyamekye@ucc.edu.gh

## Abstract

Despite its significant role in the development of society, the teaching profession is arguably one of the least preferred professions in Ghana. It has been argued that the declining interest in the teaching profession is partly contingent on the deteriorated conditions of service and the unfavourable reputation it has earned in Ghana in contemporary times. Amid these concerns, we were compelled to investigate prospective language teachers' behavioural intention to pursue a career in teaching. Using Ajzen's theory of planned behaviour, 111 prospective language teachers in two higher educational institutions in Ghana were engaged in this inquiry. The study's results, obtained through a partial least squares structural equation modelling analysis, indicated that prospective language teachers' behavioural beliefs, control beliefs, and subjective norms significantly influenced their intention to pursue a teaching career ($R^2 = 56.1\%$). We, thus, recommend that the Ministry of Education should consider addressing teachers' working conditions and launching public awareness campaigns to enhance the perception of teaching's importance. By improving teachers' professional identity, policymakers, particularly the Ghana Education Service, can attract prospective teachers and instill a positive view of teaching as an ideal profession for the youth.

## 1 Introduction

The role of teachers in the development of a society cannot be overemphasised, mainly because they are one of the main agents of society particularly tasked with transmitting the culture, history, and overall aspiration of a society to its generation [1,2]. Despite the significant role of teachers in the development of society, there seem to be concerns about the mismatch between their work input and the level

**Data availability statement:** The data underlying the results presented in the study are available from https://data.mendeley.com/datasets/4j8y23wv2b/1

**Funding:** The author(s) received no specific funding for this work.

**Competing interests:** The authors have declared that no competing interests exist.

of satisfaction with their financial returns and other related motivational supports, especially in the African educational context. Scholars contend that, though many Ghanaian teachers perceive teaching as their ideal profession, they are likely to change for a better job when the opportunity arises [3–5]. It, thus, seems that the majority of Ghanaian teachers are working as professional teachers with little motivation and are, most importantly, searching for alternative opportunities elsewhere. The foregoing could, possibly, have a detrimental effect on their instructional output, the overall reputation of the teaching profession in Ghana, as well as teacher retention [6–8].

It is worth noting that teaching as a professional career is persistently declining in its value as a reputable career opportunity in Ghana and other African countries alike [9]. Relevant literature on the teaching profession in Ghana indicates that professional teachers are always laughing stocks, as people consider the profession another means of becoming poor. Forson et al., [10] show that teaching as a profession is usually not considered a permanent career opportunity; it is rather considered a launch pad for other better career opportunities. Evidence from the literature suggests there is a high teacher attrition rate in Ghana [9,11]. Several studies have underscored the fact that Ghanaians with professional teaching backgrounds constantly seek to leave the teaching profession in search of jobs with better conditions of service, while the remaining teachers have lowered their commitment to the profession [12,13]. As suggested by Adusei et al., [12], teachers who stay in the teaching profession in Ghana probably do so because there are limited alternative job opportunities for them. The foregoing suggests that in the educational context of Ghana, teachers' professional identity remains questionable. The negative evaluation of teachers about their identity, according to Butakor, Guo [14], significantly impacts their engagement with their work, which in turn influences the academic success of their students.

Apart from the potential repercussions of in-service teachers' unfavourable evaluations of the teaching profession [15] on their instructional engagements and students' academic excellence, we assume in the context of this study that such evaluations and the reputation surrounding the professional identity of teachers could affect the willingness of prospective teachers to become professional teachers after their professional teacher education programmes in higher education. This is a gap in the literature that has not received enough scholarly attention. Thus, the current study aims to fill this scholarly void by investigating the willingness of prospective teachers to become professional teachers. We particularly seek to do this investigation using the theory of planned behaviour (TPB) as a theoretical lens. Using the TPB, we specifically seek to investigate (1) how prospective teachers' attitudes (behavioural beliefs) towards teaching influence their willingness to become professional teachers, (2) how social pressure and approvals of significant others (subjective norms) influence prospective teachers' willingness to become professional teachers, and (3) how the sense of efficacy in teaching (control beliefs) influences their willingness to become professional teachers.

We organised the paper as follows: Section 1, the introductory section, presents the background issues and, most importantly, defines the research problem and scholarly gaps that warranted the conduct of the current study. Section 2, on the other hand, reviews relevant literature. It particularly presents a detailed explication of the TPB, which serves as a theoretical lens for the study. The section also conceptualises the conceptual model proposed to test the hypotheses set to answer the set objectives of the study. Section 3 focuses on the research methods, with particular emphasis on the research design, selection of research participants, instrumentation, and how ethical issues were addressed. Section four presents and analyses the research results, while sections 5 and 6 discuss the findings and the study's implications, respectively.

## 2 Literature review

### 2.1 Theoretical framework: the theory of planned behaviour

According to Ajzen [16], understanding the subtleties of human behaviour is a difficult task. Ajzen asserts that behaviour can be measured from varying perspectives, from physiological processes to the effects of social institutions. Ajzen's theoretical framework has emerged as one of the most prominent theories commonly used in the evaluation of human behaviour, recognising and navigating its inherent complexities. The TPB assesses three crucial factors that impact a person's intention to participate in a particular behaviour. Control beliefs (an individual's perceived capability over the performance of the behaviour), subjective norms (an approval of the behaviour by significant others), and behavioural beliefs (an individual's attitude towards the behaviour) are considered the antecedents of an individual's intention. According to the TPB paradigm, every behaviour is dependent on salient information or beliefs that include normative, control, and behavioural aspects. According to TPB's theory, these beliefs become significant factors influencing a person's intentions and subsequent behaviour in a particular situation.

Fig. 1, depicted below, serves as an illustrative representation depicting the interplay of the various constructs within Ajzen's TPB. It shows the predictive capacity of the TPB constructs in discerning an individual's intention to undertake a particular behaviour.

The basic structural assumption of the TPB is that intentions are the direct cause of actions, as seen in Fig 1. However, a person's propensity to participate in a specific behaviour is heavily influenced by their attitudes toward the behaviours, subjective norms, and their perception of the behaviour's effectiveness. Perceived control beliefs affect actual behaviour both directly and indirectly. The indirect impact of control beliefs, mediated by intention, is based on the assumption that people who are effective at carrying out a particular task are more likely to have positive intentions to carry out the associated behaviours. Put simply, feeling more confident that one has the necessary skills, opportunities, and resources improves one's perception of control over behaviours. Self-efficacy beliefs, which represent an individual's perceived control over behaviours, are noteworthy because they are the only construct that can have a direct influence on behaviours execution. [16,17].

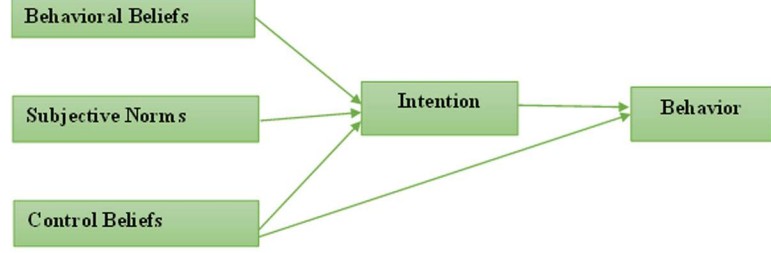

**Fig. 1. Azjen's theoretical model.**

## 2.2 Conceptual model and hypotheses development

Drawing on the theory of planned behaviour, our current study aims to investigate how prospective teachers' behavioural beliefs about professional teaching, the subjective norms about professional teaching, as well as their control beliefs, significantly contribute to their behavioural intention to become professional teachers after their teacher training education at the university. It is, thus, considered a necessity to conceptualise all these constructs, taking into consideration current research findings on how these constructs predict individuals' actions in various domains.

Behavioural beliefs: behavioural beliefs, which relate to people's attitudes towards a behaviour, constitute an essential component of the TPB, as already discussed. They represent an individual's subjective evaluation of the positive or negative outcomes associated with performing a particular behaviour. In the context of this study, behavioural beliefs are used to refer to prospective teachers' subjective evaluation of the relative importance of becoming a professional teacher. This assessment is deemed a necessity because previous literature has confirmed Ajzen's position that a positive evaluation of a particular behaviour is likely to result in an individual's intention to execute or perform the behaviour. In an investigation of teachers' attitudes towards the practice of inclusive education, for instance, Yan and Sin [18] found that, among the significant predictors of teachers' readiness to practice inclusivity in instructional activities, their attitudes (behavioural beliefs) towards inclusive education. This suggested that teachers who had favourable attitudes towards inclusive education were more willing to practice it in their teaching. Similarly, Qin and Tao [19] have confirmed that among the determinants of preservice music teachers' intention to remain in the teaching profession was the attitude they had towards the profession. These findings show that these preservice teachers were likely to opt out of the teaching profession if they had a quite unfavourable attitude towards it. Despite the existence of several studies that have applied the TPB theory to examine how attitude contributes to intention in various topics like the use of mobile learning among distance education students [20], senior high school students' willingness to use a condom [21], students' entrepreneurial intentions [22], higher education students' intention to use AI in academic writing [23], university students' intention to learn Ghanaian languages [24] among others, there seems to be a dearth of literature on how prospective teachers' attitude serves as an antecedent of their intention to become professional teachers [22]. Hence, the current study is set out to test the following hypothesis:

H1: Prospective teachers' behavioural beliefs about teaching will influence their behavioural intention to become professional teachers.

Control beliefs: It has also been established, based on the TPB, that an individual's perception of the relative difficulty or ease regarding the performance of a particular behaviour is considered a significant determinant of behavioural intention [16,17]. As Bandura [25] emphasises, teachers perceive confidence regarding their ability to execute a particular instructional activity highly correlates with the extent to which they execute that activity. This theoretical assumption has therefore compelled educational researchers to investigate the role of teacher self-efficacy in predicting classroom practices. Extant studies in Ghana have focused on understanding the role of teachers' perceived sense of efficacy concerning the enactment of particular instructional activities [26,27]. However, little is known about how prospective teachers' self-efficacy beliefs (control beliefs) contribute to their willingness to become professional teachers, hence the following hypothesis:

H2: Control beliefs about teaching will influence prospective teachers' behavioural intention to become professional teachers.

Subjective norms: This has to do with the influence of 'significant others' in shaping individuals' beliefs about the performance of a particular behaviour. It implies that people are likely to develop favourable intentions to perform a particular behaviour when people approve of their effort or intention to perform that behaviour [18]. In the context of teaching, it appears that subjective norms—i.e., public perceptions of the teaching profession are quite unfavourable, as teaching is usually viewed as an unattractive professional career by people not only in the African context but in Western cultures

as well [28]. There is evidence in Ghana that shows that even professional teachers have unfavourable perceptions of teaching as a professional career [15]. Despite little attention on the topic, there seems to be anecdotal evidence of public disapproval of teaching as an ideal profession for the youth; no wonder Forson et al., [10] consider teaching a launch pad for better opportunities for the youth. It is therefore assumed, based on the theoretical assumptions of TPB, that these social pressures and public perceptions about the status of teaching as an ideal profession can have a significant influence on prospective teachers' behavioural intentions to become professional teachers. On this basis, we test the following hypothesis:

H3: Subjective norms about the teaching profession will influence prospective teachers' behavioural intentions to become professional teachers.

Illustrated in Fig. 2 is the conceptual model depicting the proposed hypotheses and their structural linkages.

This proposed model omits the actual behaviour part of the original Ajzen's TPB model because, in this study, we dealt with prospective teachers who are yet to make their professional career choices after university education.

## 2.3 Studies on prospective teachers' behavioural intention towards teaching in Africa

Several studies in Ghana and African countries have been conducted on the determinants of career choices. Most of these studies are focused on understanding the determinants of individuals' intention to pursue a career in teaching. In Nigeria for instance, Archibong, Idaka [29] investigated the intention to pursue a career in the teaching profession among 233 final-year teacher-trainees. Using the student-teacher career questionnaire for data, the author discovered that, 135(57.9%) of the students desired to become professional teachers after graduation. Students who expressed a limited desire to pursue a career in teaching cited low societal regard for teachers, low remuneration, and the existence of out-of-field teachers in the teaching profession as factors that inhibited their behavioural intentions. The findings of Obiagu [30] provide more support on the state of preservice teachers' behavioural intentions to pursue a career in teaching. In his study, he reiterated the fact that the majority of preservice teachers in negative and professional identities which tend to affect the intrinsic desire to pursue a career in teaching.

In Tanzania, a relevant study that aimed to discover student-teachers' commitment and willingness towards becoming professional teachers was undertaken by Moses et al., [31]. After interviewing 37 undergraduate student-teachers from the Dar es Salaam University College of Education in Tanzania, these scholars concluded that some of the students were 'committed compromisers', which they explain as students who reluctantly choose to teach after school though they wish they had other opportunities. Their decision to accept teaching reluctantly was contingent on the unfavourable reputation of the teaching profession and poor working conditions. Despite the lack of intrinsic motivation among these students, they expressed the view that their decision to choose the teaching profession was influenced by the pressures of 'significant others', which is usually perceived as subjective norms in the TPB.

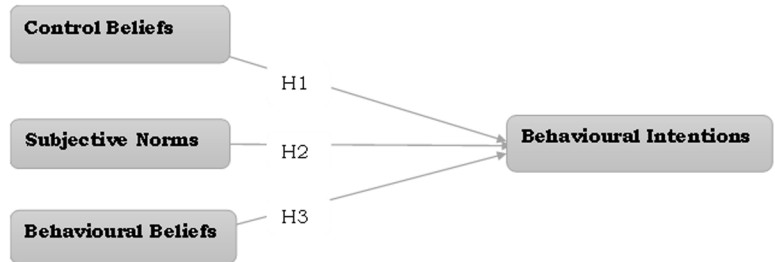

**Fig. 2. Conceptual model.**

In Ghana, there appears to be a dearth of literature particularly investigating preservice teachers' behavioural intentions to pursue a career in teaching. Nonetheless, Salifu et al., [32] conducted a study on factors that influenced pre-tertiary teachers' career choices in teaching in Ghana. Using descriptive and inferential statistics, these scholars analyzed data gathered from 354 teachers from the Greater Accra Region in Ghana. The result of their study was that the personal and social utility value of teaching was the predominant factor that influenced their choice to pursue a career in teaching. The teachers did not appreciate the influence of extrinsic utility value as a factor that influenced their career choice. Given the dearth of literature on student-teachers' behavioural intentions to pursue a career in teaching in Ghana, the current study aims to expand our understanding in this regard.

## 3 Methods and materials

### 3.1 Study design and selection of participants

A cross-sectional descriptive survey design was adopted for this study. The design was adopted because the goal of the researchers was to collect research data at a particular point in time, and importantly, describe the opinions of respondents and make generalisations [33,34]. This survey was conducted at two Ghanaian teacher development tertiary institutions. Particularly, prospective teachers pursuing Ghanaian language teaching programmes were considered for the survey. The study considered only these schools because they are the only higher educational institutions in Ghana that produce professional teachers of the local languages. The total number of prospective teachers in School A was 64 who studied either Akan, Ewe, or G. School B, on the other hand, had a total of 135 final-year prospective language teachers who studied either Ewe, Ga, Dangme, Fante, Nzema, Twi, Dagaare, Dagbani, Gonja, Gurene, Kasem, or Kusaal. Thus, in total, the population of the study was 195.

The census survey method was used for the recruitment of participants for the current study. The census method becomes an appropriate participant recruitment approach when information is to be gathered from every member of a given population, rather than selecting a sample. This method ensures complete coverage and accuracy by including all individuals in the target group [35,36]. The justification for using the census method for this study is that the total number of Ghanaian language learners in the two institutions is relatively small as scholars [35,36] recommend, conducting a census, in this case, would be feasible and practical and, most importantly, provide a more reliable dataset as it accounts for every learner, reducing errors that may arise from random sampling.

Using the census method, all 195 respondents were given a questionnaire to fill out, however, only 111 of the final-year prospective language teachers from the two traditional language teacher development tertiary institutions successfully returned the questionnaire. Hence, we had a return rate of 56.92%. However, according to the sample determination requirement of Krejcie and Morgan [37], this number of participants is representative of the entire population and could, therefore, aid generalisation

The study particularly focused on final-year prospective language teachers because these participants are on the brink of becoming professional teachers. Most importantly, we considered only the final-year prospective teachers because they had completed their teaching practicum and were aware of the experiences related to practical teaching, unlike the first-, second-, and third-year students who have not received any practical teaching experience. Thus, the final year perspective can evaluate the sense of confidence related to practical teaching, hence making them suitable candidates for this study.

### 3.2 Instrumentation and data collection procedures

The research data were gathered using an adapted TPB scale. We developed a questionnaire based on the TPB constructs to gather quantitative data. The questionnaire was composed of two major sections [Sections A and B]. The first section was meant to gather demographic information about the research participants, while the second section of the

questionnaire was a 5-point Likert scale of agreement items (ranging from strongly disagree to strongly agree) set out to measure behavioural beliefs (BB), control beliefs (CB), subjective norms (SN), and behavioural intentions (BI). To ensure the content validity of the developed research instrument, experts from the Foundations of Education at the University of Cape Coast, Ghana, were invited to scrutinise the items developed to measure the above constructs. These experts have expertise in measurement and evaluation, and thus their evaluation of the content was deemed a necessity. Moreover, statistical techniques (i.e., PLS-SEM measurement model evaluation) were used to ensure other validity requirements, such as internal consistency, convergent validity, and discriminant validity (see Section 4).

### 3.3 Data processing and analysis

Foremost, the entry of the quantitative data was aided by the Microsoft Excel sheet. After the data entry and cleaning, the data were transported to the Statistical Product and Service Solution (SPSS) version 25 for a descriptive analysis of the demographic characteristics of the research participants. However, the partial least squares structural equation modelling (PLS-SEM) analysis was aided by the SMARTPLS 4 statistical software. To test the hypotheses set for the study, we first run the measurement model analysis to confirm the reliability and validity of the TPB items. Specifically, Cronbach's alpha and the composite reliability (both RHO A and RHO C) were evaluated to measure the internal consistency of the indicator variables. The convergent validity of the indicators was also evaluated using the average variance extracted (AVE). To examine the distinctiveness of the constructs in the proposed conceptual model, the researcher used both the Fornell-Larcker criterion and the heterotrait-monotrait (HTMT) discriminant validity methods. The structural model, which aimed to test the set hypotheses of the study, was assessed using the PLS-SEM bootstrapping method with 5000 subsamples, as suggested by Hair, Risher [38].

### 3.4 Ethical consideration

Since this study involved human participants, the procedures detailed in this section were implemented to comply with the research ethics guidelines of the University of Cape Coast's institutional review board. After obtaining the research's ethical approval, questionnaires were administered to the research participants on Friday, 5th of January 2024 and ended on Sunday, 25th of February, 2024. Before data collection, students were duly informed of the purpose of the study. A written consent form was administered for participants to sign, confirming their willingness to participate in the survey. Notably, participants were assured that the information shared during the study would be treated confidentially, and they willingly engaged in the survey.

## 4  Presentation of results

### 4.1  Demographic composition of the respondents

Table 1 below presents the research participants' demographic background, taking into consideration their gender and age range.

**Table 1. Research participants' demographic background.**

|  |  | Count | % |
|---|---|---|---|
| Gender | Male | 27 | 24.3% |
|  | female | 84 | 75.7% |
| Age | 10-19 | 16 | 14.4% |
|  | 20-29 | 89 | 80.2% |
|  | 30-39 | 6 | 5.4% |
|  | 40 and above | 0 | 0.0% |

As can be observed in Table 1, a total of 111 final-year prospective language teachers took part in the study. Of these participants, 27 identified as males, while 84 identified as females. No other gender identities were disclosed. In terms of age distribution, 14.4% of the participants were within the age range of 10–19, 80.2% were within the range of 20–29, and 5.4% were within the range of 30–39. None of the participants was 40 or older.

## 4.2 Findings

### 4.2.1 Measurement model assessment.

**Model reliability and convergent validity.** Foremost, there is a need to examine the indicator loadings of each latent variable. Loadings of 0.7 or better are recommended. However, indicator variables that load as low as 0.4 could be maintained if they do not significantly affect other reliability and validity issues like the AVE [39,40]. Hence, as can be observed in Table 2 and Fig. 3, the current model indicators are loaded as recommended.

To ensure the reliability of the model, Cronbach's alpha (α), and the composite reliability (rho_A and rho_c) were evaluated in the PLS-SEM measurement model analysis (see Table 2). As Hair et al.[38] suggest, a model is said to have achieved the requisite reliability if the α, rho_a and rho_c are above the recommended threshold of 0.7. Thus, given the internal consistency index of ATT (α=.828; rho_a = 0.837; rho_c = 0.880), BI (α=.898; rho_a = 0.904; rho_c = 0.924), CB (α=.860; rho_a = 0.876; rho_c = 0.906), SN (α=.806; rho_a = 0.883; rho_c = 0.870), and CB (α=.870; rho_a = 0.876; rho_c = 0.906), we can conclude that the current model has achieved a commendable level of reliability.

**Table 2. Internal consistency, and convergent validity.**

| Constructs | Loadings | α | rho_a | rho_c | AVE |
|---|---|---|---|---|---|
| ATT | | 0.828 | 0.837 | 0.880 | 0.598 |
| att1 | 0.799 | | | | |
| att2 | 0.876 | | | | |
| att3 | 0.674 | | | | |
| att4 | 0.684 | | | | |
| att5 | 0.813 | | | | |
| BI | | 0.898 | 0.904 | 0.924 | 0.709 |
| int1 | 0.866 | | | | |
| int2 | 0.822 | | | | |
| int3 | 0.815 | | | | |
| int4 | 0.890 | | | | |
| int5 | 0.814 | | | | |
| CB | | 0.870 | 0.876 | 0.906 | 0.658 |
| cb1 | 0.727 | | | | |
| cb2 | 0.863 | | | | |
| cb3 | 0.814 | | | | |
| cb4 | 0.809 | | | | |
| cb6 | 0.837 | | | | |
| SN | | 0.806 | 0.883 | 0.870 | 0.629 |
| sn1 | 0.823 | | | | |
| sn2 | 0.768 | | | | |
| sn3 | 0.882 | | | | |
| sn4 | 0.685 | | | | |

Another significant consideration in evaluating the validity of the model is to assess the convergent validity of the model using the AVE. This assessment is considered a necessity and a prerequisite in PLS-SEM analysis because it assesses how a group of indicator variables converge to explain the variability in their underlying construct, as explained by Hair et al. [39]. The recommended AVE value necessary for the establishment of convergent validity is 0.5 or better. Hence, judging the AVE values of ATT (0.598), BI (0.709), CB (0.658), and SN (0.629), we confirm the presence of convergent validity for all of the model's constructs.

**Discriminant validity:** Discriminant validity assessment was also conducted in this study to examine how the various constructs in the model were theoretically distinct from each other. We considered this assessment very important because it is necessary to ensure that the variables in the hypothesised model are unique and are not measuring the same thing. This is important because it enhances the credibility of the findings [41,42]. Two major discriminant validity assessments were considered: the Fornell-Larcker criterion and the HTMT ratio.

HTMT ratio: Presented in Table 3 is the HTMT ratio for establishing discriminant. This procedure involves measuring the relative strength of correlation between indicators of different constructs and comparing that with the strength of

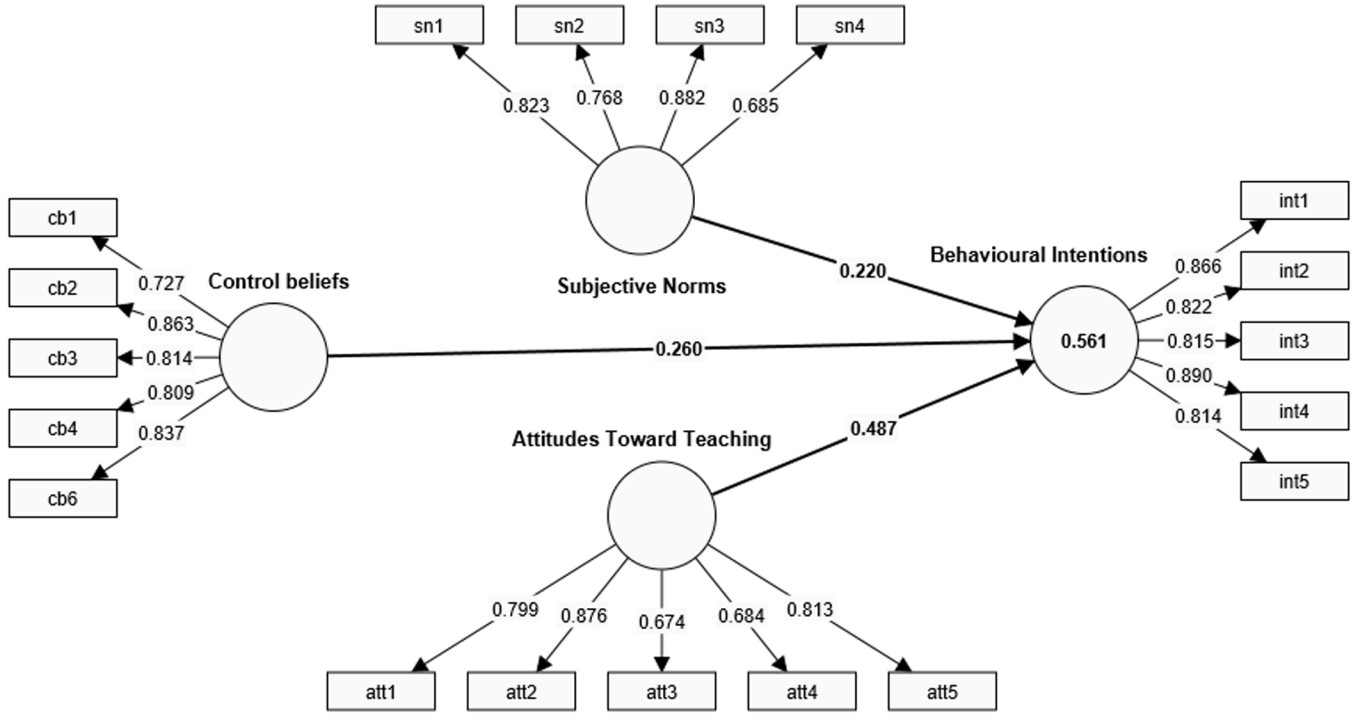

**Fig. 3. PLS-SEM measurement model.**

**Table 3. HTMT ratio.**

| Constructs | ATT | BI | CB | SN |
|---|---|---|---|---|
| ATT | | | | |
| BI | 0.761 | | | |
| CB | 0.561 | 0.583 | | |
| SN | 0.344 | 0.472 | 0.225 | |

correlation between indicators of the same construct [42]. An HTMT coefficient that approximates 1 is an indication of insufficient discriminant validity. It is usually recommended that HTMT values lie below 0.85 or 0.90. In this respect, we conclude that the constructs in the current model are below the recommended threshold, as can be seen in Table 3.

Fornell-Larcker Criterion. This criterion also provided further evidence, which indicates the presence of discriminant validity in this study's model. According to Hair et al. [39], the Fornell-Larcker criterion measures the theoretical distinctiveness of the constructs in a proposed model by comparing the square root of the AVE of a particular construct with the correlation between the constructs and other variables. It is recommended that the square root of a construct's AVE be higher than its squared correlation with other constructs. Hence, it could be concluded, based on the values presented in Table 4, that the constructs are theoretically distinct, and are thus, not measuring the same underlying concept. Multicollinearity assessment: Testing hypotheses requires an assessment of the unique predictive contribution of the predictor variable to the dependent variable (the endogenous variable). There is therefore a need for an assessment of collinearity in the proposed model. This consideration is deemed significant in dealing with misleading results due to the high correlation among the predictor variables. To address potential issues of multicollinearity, Ringle et al., [43] suggest that the variance inflation factor (VIF) of the predictors must fall below the threshold of 5. This suggests that there are no collinearity issues in the current model, as could be observed in Table 5.

**4.2.2 Structural model assessment.** The goal of the structural model assessment was to test the hypothesised relationships between the exogenous and the endogenous variables. As indicated in the introductory section, we tested how prospective teachers' behavioural beliefs, control beliefs, and subjective norms influenced their behavioural intention to become professional teachers. This was done using the PLS-SEM bootstrapping method with 5000 subsamples. The results are presented in Table 6 and illustrated in Fig. 4.

**Table 4. Fornell-Larcker Criterion.**

| Constructs | ATT | BI | CB | SN |
|---|---|---|---|---|
| ATT | 0.773 | | | |
| BI | 0.678 | 0.842 | | |
| CB | 0.477 | 0.536 | 0.811 | |
| SN | 0.303 | 0.419 | 0.200 | 0.793 |

**Table 5. Multicollinearity of predictors.**

| Constructs | ATT | BI | CB | SN |
|---|---|---|---|---|
| ATT | | 1.374 | | |
| BI | | | | |
| CB | | 1.300 | | |
| SN | | 1.106 | | |

**Table 6. PLS-SEM bootstrapping results.**

| H | Paths | β | SD | t | p | f2 | 2.5% | 97.5% | Decision |
|---|---|---|---|---|---|---|---|---|---|
| H1 | ATT ->BI | 0.487 | 0.082 | 5.924 | <0.001 | 0.394 | 0.321 | 0.639 | Supported |
| H2 | CB ->BI | 0.260 | 0.079 | 3.279 | 0.001 | 0.118 | 0.121 | 0.436 | Supported |
| H3 | SN ->BI | 0.220 | 0.075 | 2.941 | 0.003 | 0.100 | 0.079 | 0.372 | Supported |

Key: H (hypothesis); β (path coefficient); t (t values); f2 (effect size); SD (standard deviation).

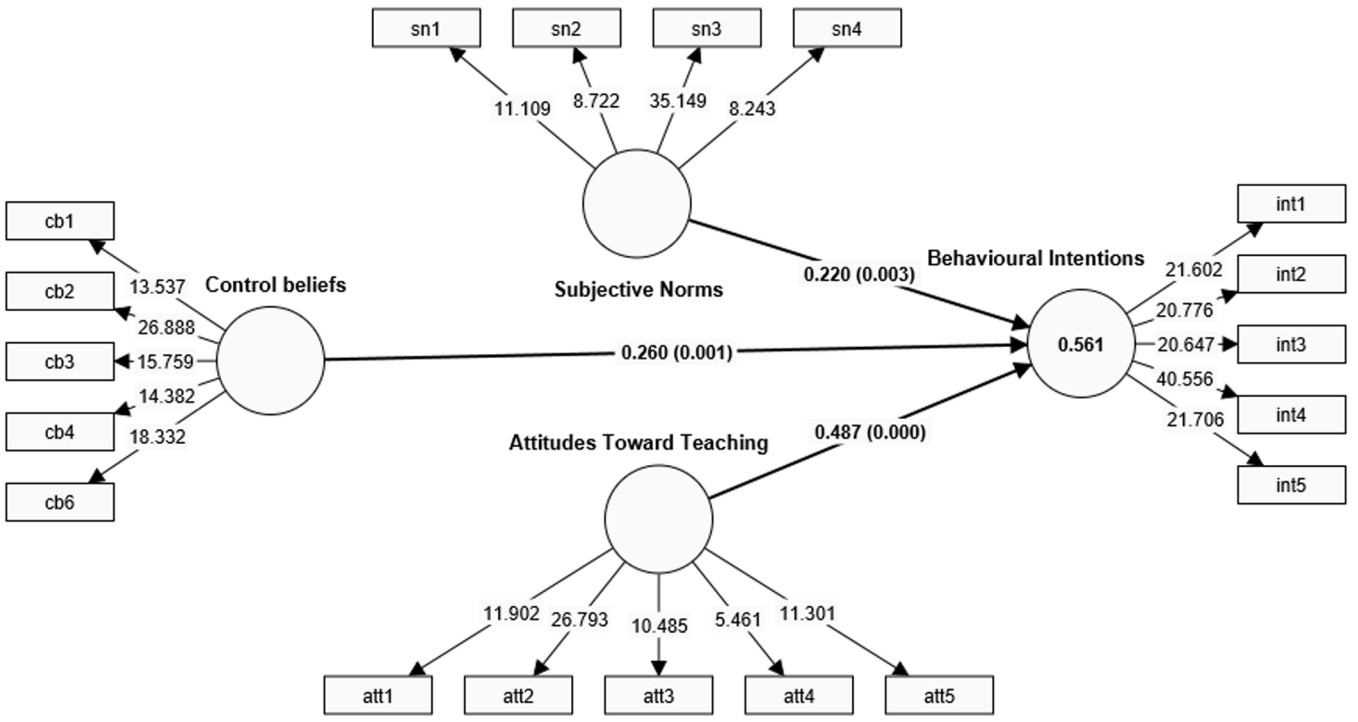

**Fig. 4. Structural model results.**

The results presented in Table 6 show that all hypotheses set in the proposed model are confirmed; thus, it could be concluded that prospective language teachers' behavioural beliefs, control beliefs, and subjective norms significantly contribute to their behavioural intention to become professional teachers. Specifically, the results show that BB has a significant positive influence on prospective teachers' behavioural intention to venture into professional teaching ($\beta = 0.487$; $t = 5.924$; $p < 0.001$; $f2 = 0.394$), hence H1 is supported. This signifies that prospective teachers are likely to choose teaching as their ideal profession as they develop a more positive attitude towards the profession. It also follows that prospective teachers who develop a relatively unfavourable attitude towards the teaching profession are likely to look for other professional opportunities. The f2 value of this hypothesised relationship shows that the impact of these participants' attitudes on their behavioural intentions is quite extreme. As Cohen [44] suggests, an effect size of 0.02, 0.15 or larger, and 0.35 or better is an indication of a small, moderate, and large effect size. This benchmark supports the assertion that attitude, in the context of this study, had a strong influence on intention.

The second hypothesis (H2), which examined the influence of prospective teachers' CB about teaching on their behavioural intentions towards becoming professional teachers, was found to be significant ($\beta = 0.260$; $t = 3.279$; $p = 0.001$; $f2 = 0.118$). The significant positive relationship established for this hypothesis is an indication that the behavioural intention magnifies as their sense of confidence and efficacy regarding teaching grows. This result implies that teachers who perceive teaching as an easy-to-do job tend to have a favourable behavioural intention to become professional teachers compared to those who have less confidence regarding the ability to teach successfully. The f2 obtained for this hypothesised relationship was moderate. This implies that, unlike BB, CB does not seem to have a strong impact on their intentions.

The result has also shown that subjective norms (SN) have a significant positive influence on prospective teachers' behavioural intention to become professional teachers ($\beta = 0.220$; $t = 2.941$; $p = 0.003$; $f2 = 0.100$), thus confirming the third

research hypothesis (H3). The insight provided by this result is that prospective teachers are likely to choose teaching as their professional career when they perceive public pressure and approvals as favourable. In other words, the subjective opinions of Ghanaians above the status of the teaching profession could compel prospective teachers to either choose to become professional teachers or seek other career opportunities. The effect size for this result is moderate, suggesting that the influence of significant 'other' has a moderate impact on the behavioural intentions of prospective teachers.

Overall, the coefficient of determination ($R^2$) shows that among a host of potential factors that could influence prospective teachers' willingness to choose teaching as a professional job, only behavioural beliefs, control beliefs, and subjection accounted for 56.1% of the proportion of variance in their behavioural intention (BI). This suggests that other factors that were not included in the current model could probably explain the remaining 43.9% of the variability in the exogenous variable. According to Hair et al. [39], $R^2$ values of approximately 0.75, 0.50, or 0.25 are substantial, moderate, and small, respectively. Hence, we conclude that a moderate proportion of the variance in the prospective teachers' behavioural intentions was explained by the exogenous variables.

In addition to the $R^2$, the predictive relevance ($Q^2$) of the model was assessed using the blindfolding method to substantiate the predictive quality of the model. Scholars contend that a model is said to have achieved predictive relevance when the $Q^2$ value exceeds 0. Specifically, a $Q^2$ value of approximately 0, 0.25, and 0.50 indicates a small, moderate, and large predictive. The current model achieved a $Q^2$ value of 0.51, which indicates that the model has a large predictive relevance.

## 5 Discussion of results

The professional identity of teachers as well as the status of the teaching profession have been a subject of scholarly debate, with most scholars arguing that Ghanaians do not seem to regard teaching as one of the most attractive professions in Ghana [15]. It has been shown, for instance, that, Ghanaian teachers' professional identity is a great threat since teachers are torn between passion and frustration. Despite some teachers' commitment to teaching, there have been concerns about losing this commitment because their constant demonstration does not seem to match up with their conditions of service [45]. The current study has provided novel insights into how these public concerns and existing studies about the professional identity of teachers as well as the status of the teaching profession contribute to prospective teachers' willingness to venture into teaching in Ghana.

From a theoretical and empirical perspective, the current study buttresses previous scholarly debates that, when given the slightest opportunity, most Ghanaian teachers are likely to quit the teaching profession for other opportunities. This study has shown that prospective teachers' attitudes towards teaching appear to be the strongest factor that contributes to their willingness to become teachers. This revelation is quite threatening to the teaching profession because prospective teachers who develop negative attitudes towards teaching may not choose it as their ideal profession. Given the negative attributes of teaching in Ghana, especially the famous conception of teaching being a 'poor man's job', coupled with the perceived bad teachers' conditions of service [46,47], and most importantly, teachers' perception of limited respect and dignity for teachers [45], prospective teachers are likely to develop an unfavourable attitude towards teaching, which may, in turn, affect their intention. There is also a possibility that the growing rate of unemployment [48,49] might compel prospective teachers to accept the teaching profession even if they possess a negative attitude towards the profession. This has a disastrous effect on productivity. In line with Ajzen's theory, prospective teachers who may venture into teaching with a negative attitude are likely to demonstrate a lackadaisical attitude towards work, especially because they may be using teaching as a launch pad for greener opportunities, as Forson et al., [10] have noticed about some profession teachers.

Closely related to attitude, are the subjective norms about teaching in Ghana. Ayinselya [45] has shown that one of the major reasons individuals entered into teaching was through the approval of their former teachers, who served as role models. Thus, their decision to become teachers was largely a function of the attractive impression their models created about the teaching profession. Ayinselya further shows that most teachers made such a career decision due to

the pressures imposed on them by their immediate family. It is, therefore, not surprising that, in the context of this study, subjective norms were shown to be a significant predictor of prospective teachers' behavioural intention to become professionals. Nonetheless, these revelations call for concerns., particularly because ideally, public perceptions of teaching are not always conducive [48]. Thus, prospective teachers' intention to wholeheartedly accept teaching remains questionable as their ideal profession amid the unfavourable social norms in Ghana [3,4,15]

Apart from the significant contribution of attitude and subjective norms to prospective teachers' behaviour and intention to become professional teachers, the study has also revealed that teachers' behavioural control—i.e., their perceived sense of confidence—regarding teaching also significantly contributed to their willingness to become professional teachers. The significant association between behavioural control and prospective teachers' intentions has shown that prospective teachers who demonstrate a stronger sense of confidence are likely to choose to teach, while those with less confidence might contemplate whether or not they would succeed as professional teachers. This substantiates the theoretical assumption of Bandura [25], which underscores the role of teachers' self-efficacy beliefs in their teachers' professional success.

## 6 Implications of the study

### 6.1 Theoretical implications

The findings of the study have supported the predictive potential of Ajzen's theory of planned behaviour. In the context of this study, the TPB theory has shown how attitude, subjective norms, and control beliefs explain teachers' intentions towards becoming professionals. With this theoretical lens, we have been able to discuss and draw conclusions that prospective teachers' willingness to step into professional teaching could be, among other significant factors, a function of the attitude they have developed about the teaching profession, the subjective norms surrounding the teaching profession and the professional identity of teachers, and their sense of efficacy regarding teaching. Overall, the empirical evidence of this study provides support for the theoretical validity of the TPB.

### 6.2 Managerial implications

Though the study was conducted in a specific context of language teaching, which does not fully represent the broader context of teachers in other academic fields, the findings of the current study have implications for teachers in general. It provides insight into the career intention of prospective teachers in other fields, given that teachers' working conditions in Ghana, irrespective of the field of the teacher, are perceived to be suboptimal. The findings also have implications for any context where the profession suffers from a poor reputation as well as in contexts like Ghana and African countries alike where the government seeks to improve teacher recruitment and retention.

To counter the negative perception of teaching as a 'poor man's job', it is crucial to tackle this stereotype at its root. Educational institutions and policymakers must launch campaigns to reshape societal attitudes towards the profession. Showcasing successful teachers, improving their working conditions, and emphasising their vital role in society can all contribute to shifting the narrative. It's equally important to foster respect and dignity for teachers. Acknowledging and honouring their contributions within schools and society can significantly enhance the perception of their profession. This multifaceted approach can help address the concerns surrounding the inclination towards alternative careers in education.

## 7 Limitations and suggestions for future research

Despite the contribution of this study in shaping our understanding of the predictors of prospective teachers' behavioural intention to become professional teachers, we outline some limitations that warrant the conduct of further studies to broaden our understanding of the topic under scholarly discourse. Foremost, the quantitative approach to this investigation presents a methodological gap; we thus suggest that future studies must employ an in-depth qualitative study to

broaden understanding of issues that may shape prospective teachers' behavioural intentions. Also, the involvement of only prospective language teachers presents a population gap. Future research can, therefore, fill this void by investigating the same issue from the perspective of prospective teachers in diverse fields of study and educational institutions in Ghana. Another limitation of this study is that it did not include teachers who are already in the field delivering their professional duties. To this end, the actual behavioural aspect was omitted from Ajzen's TPB model in this study because prospective teachers are yet to actually be recruited, and for that matter, their actual behaviour could not be measured. Therefore, we cannot guarantee that if they are actually recruited, their intentions to actually execute their professional mandates or not will be affected or reflect their responses in this study. To this end, future studies should include professional teachers who are already employed and delivering their duties, so as to do a comparative analysis between both categories. Thus, professional teachers who are already recruited and prospective teachers so as to portray a comprehensive understanding of how individuals' behavioural intentions are influenced by behavioural beliefs, subjective norms, and control beliefs. Finally, there is a need for future research to investigate how prospective teachers' behavioural intentions could be moderated by demographic characteristics, most importantly, gender, and family financial background.

## Supporting information

**S1.** Questionnaire.docx.
(DOCX)

## Author contributions

**Conceptualization:** Ernest Nyamekye, John Zengulaaru.

**Investigation:** Ernest Nyamekye, John Zengulaaru.

**Methodology:** John Zengulaaru.

**Writing – review & editing:** Ernest Nyamekye, John Zengulaaru.

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
