## [Decision Letter · Decision Letter 0]

15 May 2024

PONE-D-24-09444‘Who Wants to be a Teacher in Ghana?’ A Structural Equation Modelling Analysis of Prospective Teachers’ Behavioural Intention to Pursue a Career in TeachingPLOS ONE

Dear Dr. Nyamekye,

Thank you for submitting your manuscript to PLOS ONE. After careful consideration, we feel that it has merit but does not fully meet PLOS ONE’s publication criteria as it currently stands. Therefore, we invite you to submit a revised version of the manuscript that addresses the points raised during the review process.

We look forward to receiving your revised manuscript.

Kind regards,

Ibrahim Adeshola

Academic Editor

PLOS ONE

Reviewers' comments:

Reviewer's Responses to Questions

**Comments to the Author**

1. Is the manuscript technically sound, and do the data support the conclusions?

Reviewer #1: Partly

Reviewer #2: Partly

Reviewer #3: Partly

2. Has the statistical analysis been performed appropriately and rigorously? 

Reviewer #1: Yes

Reviewer #2: Yes

Reviewer #3: No

3. Have the authors made all data underlying the findings in their manuscript fully available?

Reviewer #1: Yes

Reviewer #2: No

Reviewer #3: Yes

4. Is the manuscript presented in an intelligible fashion and written in standard English?

Reviewer #1: No

Reviewer #2: Yes

Reviewer #3: Yes

5. Review Comments to the Author

Reviewer #1: Overall, in the review of the manuscript, several areas for improvement were identified across different sections. Starting with the Introduction, reviewers noted that the section lacked a clear structure and failed to effectively convey the study's purpose and significance. I recommended revising the introduction to provide a clearer overview of the research aims and objectives, as well as to establish the context for the study more effectively. In the Literature Review, reviewers suggested strengthening the integration of existing literature to provide more robust theoretical support for the study. I advised ensuring that the review is comprehensive and well-organized, with a clear progression of ideas and a critical analysis of relevant studies.

Moving on to the Abstract, i notice that while it provided a brief summary of the study, it lacked clarity and did not follow a structured format. I recommended restructuring the abstract to include sections on clarity, methodology, findings, recommendations, and policy implications, and ensuring that each section flows logically and succinctly summarizes the key points of the study. Overall, the review aimed to provide constructive feedback to enhance the clarity, coherence, and overall effectiveness of the manuscript, from the Introduction and Literature Review to the Abstract and beyond.

Reviewer #2: PONE

Reviewer’s Comments

Title: ‘Who Wants to be a Teacher in Ghana?’ A Structural Equation Modelling Analysis of Prospective Teachers’ Behavioural Intention to Pursue a Career in Teaching

General Comments

1. The paper is carefully written and edited for language, clarity of expressions and grammar though proofreading is still required to fix a few errors.

2. The paper is concisely written avoiding unnecessary details.

3. The methodology deployed seems inadequate as some important information regarding sampling, sample size etc. are missing.

Abstracts

1. The abstract is a true reflection of the work, and it is well written.

Introduction

1. The introduction provides enough scope and a good and clear setting for the study.

2. All major themes pertaining to the study have been discussed in the introduction.

3. Moreover, the authors have explicitly showed the exact problem and motivation underlying the study.

4. However, the contributions of the study are not stated in the introduction. Even though, I expect elaborate discussion of contributions in the literature review section, brief highlight of contributions in the introduction is also useful for readers to know the contribution of a study right from its introduction section.

5. Moreover, the authors do not present any peer-reviewed or published evidence/argument(s) in support of the claim that “teaching as a professional career is persistently declining in its value as a reputable career opportunity in Ghana and other African countries alike”. I encourage authors to do this to give more structural credibility to the content of the introduction in terms of the authenticity and severity of the problem being examined. Remember to support all arguments with statistics and/or literature where necessary and available.

Literature review

1. The literature starts with some (one or two) of the initial works in the area and how they advanced and shaped research in that area. That’s great.

2. Also, there is clear logical links and various strands of thought. Authors have connected their study with the wider research literature and have consequently positioned their research appropriately.

3. Excellent empirical literature review.

4. However, it should be strengthened in the following ways:

a. The authors must highlight the salient gaps in the literature and demonstrate how they intend to contribute to filling them.

b. The authors must review a few more current empirical literatures on the specific subject of “Prospective Teachers’ Behavioural Intention to Pursue a Career in Teaching”. Even if you don’t literature related to teaching you could find some related to other professions/careers.

Methods and Materials

1. It is not clear why you chose only two schools and not say five, or ten schools. Even if the two schools chosen is enough, why would you choose say school A and B instead of C and D? While you may not have to mention the names of the schools, their selection criteria must be revealed to inform the validity of the sample. Similarly, why did you choose to focus on Ghanaian languages and not core subjects? In a nutshell, you must provide a scientific selection criterion rather than a personal or arbitrary selection criteria.

2. Again, the sampling technique employed in selecting the 111 respondents is not mentioned, let alone what led to the decision to use 111 respondents. All these must be discussed in the revised version of the manuscript.

3. I understand that you have included some of them as limitations of the study, but you still have to explain your choices because knowing the selection/choice criteria give readers a good understanding of how scientific are the methodological procedures you deployed for the study.

Results

1. Most of the findings are succinctly discussed and properly linked with the overarching objectives of the study.

2. Furthermore, most of the findings are very interesting, practical and could potentially guide policy.

3. However, the discussion could be strengthened by highlighting some plausible explanation why, for instance, subjective norms about teaching influences entry into the teaching profession. I urge authors to do the same for the remaining two factors of TPB (i.e., behavioural control— and behavioural beliefs)

4. Importantly, due to aforementioned problems with the methodology section I doubt if the results are scientific enough. I will wait for author’s response/action on the methodology to make a final evaluation of this section.

5. Otherwise, the section is well written.

Implications

1. This is well written.

2. Yet, authors could have said something about the extent to which your findings can be generalized and highlighted contexts in which they could be applicable.

References

1. Well written

Reviewer #3: The authors should check the journal’s guidelines and format their paper according. E.g. abstract and referencing

Introduction

- The introduction is poorly written. If you begin on the premise, what is the essence of your study?

- Better framing would have been using data on teacher attrition and teacher development in Ghana to justify your study.

- Then offer solid rational for studying teacher motivation.

- If they already in teacher training programme, then they have already made their mind or decided their career choice.

- Current status of teachers in Ghana could also help in justifying why we need to study teacher motivation.

Theoretical framework

TPB has been applied in studies in Ghana and thus, the authors should do a job in explaining how the tenets were operationalized for their study. Explain what the theory is about then indicate how you are going to operationalize it in your study

Conceptual model

Section 2.2 could be deleted. If you explain your model properly, you could highlight how it has been applied then come up with your hypothesis. Ajzen has already argued that the three predictors mediate through intention to predict a given behaviour. The studies you used to developed your hypothesis has nothing to do with teacher motivation! All that you hypothesized was advanced by Ajzen!

Method

The method tells me that the who study frame is misleading. If you drew on language teachers, then that should drive your who study… why did recruit language teachers? You did not study teachers’ intention however, you studied language teachers intention.

Instrument

There is lack of transparency regarding how the instruments were developed. How were they developed? How can we trust the content and its alignment to the tenets of the TPB?

We cannot trust the result until this is substantiated.

6. PLOS authors have the option to publish the peer review history of their article (what does this mean? ). If published, this will include your full peer review and any attached files.

**Do you want your identity to be public for this peer review?** For information about this choice, including consent withdrawal, please see our Privacy Policy .

Reviewer #1: **Yes: ** Dokun I. Oluwajana

Reviewer #2: **Yes: ** Solomon Aboagye

Reviewer #3: No

---

## [Author Response · Author response to Decision Letter 1]

27 May 2024

Rebuttal Letter

Abstract

Comment: Moving on to the Abstract, I notice that while it provided a brief summary of the study, it lacked clarity and did not follow a structured format. I recommended restructuring the abstract to include sections on clarity, methodology, findings, recommendations, and policy implications, and ensuring that each section flows logically and succinctly summarizes the key points of the study. Overall, the review aimed to provide constructive feedback to enhance the clarity, coherence, and overall effectiveness of the manuscript, from the Introduction and Literature Review to the Abstract and beyond.

Response: Thank you for the suggestion. The suggestion has been taking into consideration as suggested

Comment: The abstract contains a considerable amount of jargon. The phrase "Amidst declining interest in teaching" lacks clarity and coherence. It should be rephrased to form a complete sentence and maintain a logical flow throughout. I suggest restructuring the abstract to include sections on clarity, methodology, findings, recommendations, and policy implications. Further adjustments are recommended to enhance the abstract's effectiveness.

Response: Thank you for the suggestion. The suggestion has been taking into consideration as suggested

.

Introduction

Comment 1: Reviewer #1: Overall, in the review of the manuscript, several areas for improvement were identified across different sections. Starting with the Introduction, reviewers noted that the section lacked a clear structure and failed to effectively convey the study's purpose and significance. I recommended revising the introduction to provide a clearer overview of the research aims and objectives, as well as to establish the context for the study more effectively. In the Literature Review, reviewers suggested strengthening the integration of existing literature to provide more robust theoretical support for the study. I advised ensuring that the review is comprehensive and well-organized, with a clear progression of ideas and a critical analysis of relevant studies

Response: As could be seen in pages 3 and 4, we had already clearly outlined the purpose of the as follows:

“Thus, the motivation for the current study is to investigate the willingness of prospective teachers to become professional teachers. We particularly seek to do this investigation using the theory of planned behaviour (TPB) as a theoretical lens. Using the TPB, we specifically seek to investigate how (1) prospective teachers’ attitudes (behavioural beliefs) towards teaching influence their willingness to become professional teachers, (2) how social pressure and approvals of significant others (subjective norms) influence prospective teachers’ willingness to become professional teachers, and (3) how the sense of efficacy in teaching (control beliefs) influence their willingness to become professional teachers.” We therefore wonder what other purpose of the study the reviewer expects us to clearly outline. We would be glad if the reviewer could be more specific

Comment 2: Reviewer 2# However, the contributions of the study are not stated in the introduction. Even though, I expect elaborate discussion of contributions in the literature review section, brief highlight of contributions in the introduction is also useful for readers to know the contribution of a study right from its introduction section.

Response: We thank the reviewer for this contribution. However, from our standpoint, the discussion of results and implications of the study sections are the ideal sections for demonstrating how a study contributes to readers understanding of a particular topic. We do not intend to nullify the significance of the reviewer’s submission; nonetheless, per our writing style, dedicating the entire introduction to background issues that compels readers to find out what the study seeks to achieve is ideal. However, if the journal requires that the contributions of the study set the tone in the introduction, we would be ready to take that into consideration..

Comment 3: Reviewer 2# Moreover, the authors do not present any peer-reviewed or published evidence/argument(s) in support of the claim that “teaching as a professional career is persistently declining in its value as a reputable career opportunity in Ghana and other African countries alike”. I encourage authors to do this to give more structural credibility to the content of the introduction in terms of the authenticity and severity of the problem being examined. Remember to support all arguments with statistics and/or literature where necessary and available.

Response: The suggestion above has been addressed. We have provided a citation about the declining status of the teaching profession in Ghana and other African countries, as can be observed on page 3, paragraph 2.

Comment 4: Reviewer #3: The introduction is poorly written. If you begin on the premise, what is the essence of your study? Better framing would have been using data on teacher attrition and teacher development in Ghana to justify your study. Then offer solid rational for studying teacher motivation.

Response: We thank the reviewer for this suggestion. We have addressed this issue. With relevant literature, we have grounded the study on the premise that teacher attrition in Ghana is high. Moreover, the value of the teaching profession is said to be declining, according to literature. This thus made us wonder whether prospective teachers would develop a favourable behaviour

Response:

Comment 5: If they already in teacher training programme, then they have already made their mind or decided their career choice.

Response: We thank the reviewer for this submission. However, we wish to express our disagreement with this fact. The fact that these students are in the teacher training programme does not guarantee that they have made their career choices already. Theoretically, that is the expectation. However, this deviates from reality because my interactions with the students I teach in higher education in Ghana show that most of them are contemplating their career choices given the economic conditions of the country and the reputation regarding the economic condition of the profession. Moreover, taking a pursuing profession course in teaching in Ghana does not mean that graduates are limited to teaching only; they equally have the opportunity to work in other sectors of the economy as well.

Moreover, all students are, at the end of the end of their final year, given entrepreneurial education. This initiative suggests that all graduates are encouraged to venture into entrepreneurship. This also buttresses the fact that students in the context of this study are encouraged to seek other employment opportunities despite being professionally trained as teachers.

Comment 6: Current status of teachers in Ghana could also help in justifying why we need to study teacher motivation.

Response: This is already captured in the introduction. The introduction is emphatic on the fact that teachers’ status is quite unfavourable given the declining teacher reputation and the constant rant about bad conditions of service, which consequently affect teacher retention and commitment to the profession (see pages 2 and 3).

Comment 7: *The introduction lacks a strong structure and relies on assumptions rather than scientific evidence. For instance, statements like " This, from our perspective, has a detrimental effect...& quot; Anecdotal evidence and popular opinions in Ghana indicate that professional teachers are always laughing stocks, as people consider the profession …………” should be revised to present evidence-based claims.

Response: We thank the reviewer for pointing out this issue. We have addressed this issue by adding more empirical literature that supports our assumptions, as could be observed in the introduction.

Comment: The introduction contains grammatical errors, such as awkward phrasing and incorrect use of punctuation. For example, " Several studies in Ghana, for instance, have shown that, & quot; could be revised for clarity.

Response: the said correction has been made, as could be seen in the first paragraph of the introduction

Comment. *****Unclear Message - The introduction fails to effectively communicate the main message to the reader. It should provide a clear overview of the study's significance, context, and objectives.

Response: We thank the reviewer for this suggest. However, the suggestion is quite unclear, since we have already given the overview of the study as well as the objectives of the study

Literature review

Comment 1: Excellent empirical literature review. However, it should be strengthened in the following ways:

i. The authors must highlight the salient gaps in the literature and demonstrate how they intend to contribute to filling them.

Response: This has been addressed. After the discourse in the introduction, we stated as follows:

Apart from the potential repercussion of in-service teachers' unfavourable evaluation of the teaching profession (1) on their instructional engagements and students’ academic excellence, we assume in the context of this study that such evaluations and the reputation surrounding the professional identity of teachers could affect the willingness of prospective teachers to become professional teachers after their professional teacher education programmes in higher education. This appears to be a gap in the literature yet to receive enough scholarly attention. Thus, the current study aims to fill this scholarly void by investigating the willingness of prospective teachers to become professional teachers. We particularly seek to do this investigation using the theory of planned behaviour (TPB) as a theoretical lens (see second paragraph on page ).

Comment 2: The authors must review a few more current empirical literatures on the specific subject of “Prospective Teachers’ Behavioral Intention to Pursue a Career in Teaching”. Even if you don’t literature related to teaching you could find some related to other professions/careers.

Response: This has been done already, as could be seen in the conceptual model section.

Theoretical framework

Comment 1: Reviewer #3. TPB has been applied in studies in Ghana and thus, the authors should do a job in explaining how the tenets were operationalized for their study. Explain what the theory is about then indicate how you are going to operationalize it in your study

Response: We thank the reviewer for this suggestion aimed at improving the overall quality of the paper. However, this suggestion has already been dealt with under the theoretical framework and model conceptualization sections. The theoretical framework section was particularly dedicated to defining Ajzen’s TPB and its tenets. We discussed how the TPB, according to Ajzen, could be applied to understand the forces behind individuals’ actions. Under the conceptual model section, we moved on to meticulously present and discuss the TPB theory, which was used to measure prospective teachers’ behaviour regarding their willingness to pursue a career in teaching. As we clearly stated in the opening paragraph of the conceptual model section:

“Drawing on the theory of planned behaviour, our current study aims to investigate how prospective teachers’ behavioural beliefs about professional teaching, the subjective norms about professional teaching, as well as their control beliefs, significantly contribute to their behavioural intention to become professional teachers after their teacher training education at the university. It is, thus, considered a necessity to conceptualise all these constructs, taking into consideration current research findings on how these constructs predict individuals’ actions in various domains.”

Having clearly defined how the TPB was to be applied in the current study, we find it quite challenging to understand what else the reviewer expects in terms of “operationalizing the theory” in our study. All our operational hypotheses were based solely on the tenets of the theoretical framework and empirical literature. However, we would be glad if the reviewer could be more specific on the exact thing we should do that is different from what we have already done.

Comment 2: While manuscript attempts to explain TPB, some sentences are overly complex and can be simplified for clarity. For instance, phrases like "comprehensive perspective that has propelled the Theory of Planned Behaviour (TPB) to be at the forefront of theories frequently employed..." could be streamlined for easier comprehension.

Response: the entire section has been language-edited to improve clarity and comprehensibility. We thank the reviewer for pointing this out

Comment 3: The discussion of theoretical framework may benefit from being more concise. Some sentences appear repetitive or overly verbose and can be condensed without losing important information.

Response: this has been addressed (refer to response to comment 2)

Comment: The flow of ideas can be enhanced to improve coherence. Transitions between the different aspects of the TPB and its components could be smoother, allowing readers to follow the discussion more easily.

Response: Addressed

Conceptual model

Comment: Reviewer #3. Section 2.2 could be deleted. If you explain your model properly, you could highlight how it has been applied then come up with your hypothesis. Ajzen has already argued that the three predictors mediate through intention to predict a given behaviour. The studies you used to developed your hypothesis has nothing to do with teacher motivation! All that you hypothesized was advanced by Ajzen!

Response: We thank the reviewer for the suggestion. However, we wish to oppose this suggestion. Foremost, we wish to declare that there is a need to maintain the conceptual model. In applying PLS-SEM, there is a need to explain the theoretical framework that underlies the study. However, there is always a need to build a model out of the theoretical framework to show how the theory is practically used in the study. As we did in the study, we did not use all the tenets of the TPB. Thus, there is a need to adapt the TPB in a conceptual form to show how it is applied.

Also, there seems to be no established convention that the literature reviewed in a particular study must be exactly related to the study’s topic. Thus, stating that the reviews have nothing to do with teacher motivation is unacceptable from our standpoint. The main aim of the reviews under the conceptual model was to revisit studies that have used some of the tenets of the TPB to understand human behaviour. Besides, the issues discussed are within the scope of teachers and school-related issues. It does not deviate significantly from the study’s topic. Moreover, we wish to declare that the study is not even about teacher motivation; it is rather about their intention—from our perspective, intention and motivation are not the same.

Methodology

Comment: Reviewer 2#The methodology deployed seems inadequate as some important information regarding sampling, sample size etc. are missing.

Response: This has been resolved, the population and sample are been clearly defined and justified

Comment: Reviewer 2# It is not clear why you chose only two schools and not say five, or ten schools. Even if the two schools chosen is enough, why would you choose say school A and B instead of C and D? While you may not have to mention the names of the schools, their selection criteria must be revealed to inform the validity of the sample. Similarly, why did you choose to focus on Ghanaian languages and not core subjects? In a nutshell, you must provide a scientific selection criterion rather than a personal or arbitrary selection criteria.

Response: We have justified our selection criteria. We must state that our study focused solely on language teachers in two higher education institutions that specialise in training indigenous language teachers. For this reason, we have modified our research topic to limit the generality of the study. (See page 9)

Comment: Reviewer 2# The Again, the sampling technique employed in selecting the 111 respondents is not mentioned, let alone what led to the decision to use 111

---

## [Decision Letter · Decision Letter 1]

21 Jun 2024

PONE-D-24-09444R1Who wants to be a teacher in Ghana?’ a structural equation modelling analysis of prospective language teachers’ behavioural intentions to pursue a career in teachingPLOS ONE

Dear Dr. Nyamekye,

Thank you for submitting your manuscript to PLOS ONE. After careful consideration, we feel that it has merit but does not fully meet PLOS ONE’s publication criteria as it currently stands. Therefore, we invite you to submit a revised version of the manuscript that addresses the points raised during the review process.

We look forward to receiving your revised manuscript.

Kind regards,

Ibrahim Adeshola

Academic Editor

PLOS ONE

Additional Editor Comments:

Please pay attention to the constructive comments by the reviewers and make changes to the study.

Reviewers' comments:

Reviewer's Responses to Questions

**Comments to the Author**

1. If the authors have adequately addressed your comments raised in a previous round of review and you feel that this manuscript is now acceptable for publication, you may indicate that here to bypass the “Comments to the Author” section, enter your conflict of interest statement in the “Confidential to Editor” section, and submit your "Accept" recommendation.

Reviewer #1: (No Response)

Reviewer #2: (No Response)

2. Is the manuscript technically sound, and do the data support the conclusions?

Reviewer #1: No

Reviewer #2: Partly

3. Has the statistical analysis been performed appropriately and rigorously? 

Reviewer #1: Yes

Reviewer #2: Yes

4. Have the authors made all data underlying the findings in their manuscript fully available?

Reviewer #1: Yes

Reviewer #2: No

5. Is the manuscript presented in an intelligible fashion and written in standard English?

Reviewer #1: No

Reviewer #2: Yes

6. Review Comments to the Author

Reviewer #1: I have read the reviewers' comments and authors feedback. It is important that the authors carefully reviews the comments and answers the questions accordingly. Instead of the authors justifying the comments, they need to correct and modify the manuscript. Additionally, I still have concerns regarding the grammar, which does not meet the required technical standards of the journal. Thanks

Reviewer #2: PONE

Reviewer’s Comments (Revision 1)

Title: ‘Who Wants to be a Teacher in Ghana?’ A Structural Equation Modelling Analysis of Prospective Teachers’ Behavioural Intention to Pursue a Career in Teaching

General Comments

1. Some proofreading is still required to fix a few errors.

Introduction

1. Authors seems to be unsure of their contributions because they claim that “This appears to be a gap in the literature yet to receive enough scholarly attention. Gaps must be real, factual and contributions should be obvious and relevant coming directly from the identified gap(s).

Literature review

1. Excellent empirical literature review. However, it should be strengthened in the following ways:

a. The authors must highlight the salient gaps in the literature and demonstrate how they intend to contribute to filling them.

b. The authors must review a few more current empirical literatures on the specific subject of “Prospective Teachers’ Behavioural Intention to Pursue a Career in Teaching”. Even if you don’t literature related to teaching you could find some related to other professions/careers.

Methods and Materials

1. So, what was the sampling technique? Random sampling or a mere point of saturation was employed?

Results

1. The discussion could be strengthened by highlighting some plausible explanation why, for instance, subjective norms about teaching influences entry into the teaching profession. I urge authors to do the same for the remaining two factors of TPB (i.e., behavioural control— and behavioural beliefs)

Implications

1. Authors could have said something about the extent to which your findings can be generalized and highlighted contexts in which they could be applicable.

7. PLOS authors have the option to publish the peer review history of their article (what does this mean? ). If published, this will include your full peer review and any attached files.

**Do you want your identity to be public for this peer review?** For information about this choice, including consent withdrawal, please see our Privacy Policy .

Reviewer #1: No

Reviewer #2: **Yes: ** Solomon Aboagye

---

## [Author Response · Author response to Decision Letter 2]

26 Jun 2024

Response to Reviewers

Reviewer #1: I have read the reviewers' comments and authors feedback. It is important that the authors carefully reviews the comments and answers the questions accordingly. Instead of the authors justifying the comments, they need to correct and modify the manuscript. Additionally, I still have concerns regarding the grammar, which does not meet the required technical standards of the journal. Thanks

Response: the grammatical issues has been taking care of. Thank You

Reviewer #2: PONE

1. Some proofreading is still required to fix a few errors.

Response: the manuscript has been proofread.

2. Introduction

1. Authors seems to be unsure of their contributions because they claim that “This appears to be a gap in the literature yet to receive enough scholarly attention. Gaps must be real, factual and contributions should be obvious and relevant coming directly from the identified gap(s).

Response: I seems the reviewer pointed this out because of the choice of construction “This appears..” to make this clear we have rephrased the entire section making our gap emphatic as could be observed in page 4 paragraph 2

3. Literature review

1. Excellent empirical literature review. However, it should be strengthened in the following ways:

a. The authors must highlight the salient gaps in the literature and demonstrate how they intend to contribute to filling them.

Response: We wish to point out that our study focuses on filling one primary knowledge gap as could be observed in page 4 paragraph 2

b. The authors must review a few more current empirical literatures on the specific subject of “Prospective Teachers’ Behavioural Intention to Pursue a Career in Teaching”. Even if you don’t literature related to teaching you could find some related to other professions/careers.

Response: Again, it is quite challenging to identify specific studies that are closely tied to Prospective Teachers’ Behavioural Intention to Pursue a Career in Teaching. Also, getting a literature relation to prospective teachers’ behavioural intention to pursue a career in other profession would be far-fetch. We do not appreciate how that would fit into our argument and how it contributes to our research gap. If the reviewer insists on the need to review literature on this, we would appreciate it if relevant literature in this regard is provided for us. Currently, we find it difficult to identify any.

4. Methods and Materials

1. So, what was the sampling technique? Random sampling or a mere point of saturation was employed?

Response: In our methods section, we stated emphatically that the we used census method (which involves a consideration of all possible research subjects in an investigation). Hence, it is difficult to understanding why the reviewer expects us to state whether we did simple random sampling or not. We stand to be corrected anyway, but we are certain that census method is another procedure used in getting research subjects for an investigation. But considering all subjects does not necessary all of the population would be involved. As could be observed in our study, the decision was to include all members of the population, by only 180 of them respondent to the questionnaire. It was a mere point of saturation either. The sampling technique is census method (census sampling). This is one of the many studies that employed this sampling technique:

BISWAS, S., & SEHGAL, V. K. (1987). Generalized probability model for estimating the proportion of the female population at different levels of fertility based on census sampling from a mixed population. International Journal of Systems Science, 18(10), 1909–1917. https://doi.org/10.1080/00207728708967163 .

Janatolmakan, M., Andaieshgar, B., Aryan, A., Jafari, F., & Khatony, A. (2019). Comparison of Depression Rate Between the First- and Final-Year Nursing Students in Kermanshah, Iran. Psychology Research and Behavior Management, 12, 1147–1153. https://doi.org/10.2147/PRBM.S238873

Raige, S. R., Stellefson, M., Chaney, B. H., & Alber, J. M. (2015). Pinterest as a Resource for Health Information on Chronic Obstructive Pulmonary Disease (COPD): A Social Media Content Analysis. American Journal of Health Education, 46(4), 241–251. https://doi.org/10.1080/19325037.2015.1044586

5. Results

1. The discussion could be strengthened by highlighting some plausible explanation why, for instance, subjective norms about teaching influences entry into the teaching profession. I urge authors to do the same for the remaining two factors of TPB (i.e., behavioural control— and behavioural beliefs)

Response: We would like to urge the reviewer to critically have a second look at the discussion section. Our discussion touches on why the three predictors had a significant association with Intention as well as the possible as the possible implications. Probably, the reviewer does not appreciate the style in which the arguments were presented. We have highlighted those areas in red.

6. Implications

1. Authors could have said something about the extent to which your findings can be generalized and highlighted contexts in which they could be applicable.

Response: This issue has been addressed. In the managerial implication section in page 23, we have highlighted the context in which the findings could be applicable, indicating the need for stakeholders to take those contexts into considerations.

---

## [Editor Report · Decision Letter 2]

2 Jul 2024

PONE-D-24-09444R2Who wants to be a teacher in Ghana?’ a structural equation modelling analysis of prospective language teachers’ behavioural intentions to pursue a career in teachingPLOS ONE

Dear Dr. Nyamekye,

Thank you for submitting your manuscript to PLOS ONE. After careful consideration, we feel that it has merit but does not fully meet PLOS ONE’s publication criteria as it currently stands. Therefore, we invite you to submit a revised version of the manuscript that addresses the points raised during the review process.

We look forward to receiving your revised manuscript.

Kind regards,

Ibrahim Adeshola

Academic Editor

PLOS ONE

Additional Editor Comments:

Please carefully review and address the reviewers' comments accordingly.

---

## [Author Response · Author response to Decision Letter 3]

3 Jul 2024

Response to Reviewers

Reviewer #1: I have read the reviewers' comments and authors feedback. It is important that the authors carefully review the comments and answers the questions accordingly. Instead of the authors justifying the comments, they need to correct and modify the manuscript. Additionally, I still have concerns regarding the grammar, which does not meet the required technical standards of the journal. Thanks

Response: the grammatical issues has been taking care of. Thank You

Reviewer #2: PONE

1. Some proofreading is still required to fix a few errors.

Response: the manuscript has been proofread.

2. Introduction

1. Authors seems to be unsure of their contributions because they claim that “This appears to be a gap in the literature yet to receive enough scholarly attention. Gaps must be real, factual and contributions should be obvious and relevant coming directly from the identified gap(s).

Response: I seem the reviewer pointed this out because of the choice of construction “This appears...” to make this clear we have rephrased the entire section making our gap emphatic as could be observed in page 4 paragraph 2

3. Literature review

1. Excellent empirical literature review. However, it should be strengthened in the following ways:

a. The authors must highlight the salient gaps in the literature and demonstrate how they intend to contribute to filling them.

Response: We have provided a new subsection that reviews empirical studies related to student-teachers' behavioural intentions towards pursuing a career in teaching (see page 9 and 10). At the end of the section, we highlight an empirical gap the study fills.

b. The authors must review a few more current empirical literature on the specific subject of “Prospective Teachers’ Behavioral Intention to Pursue a Career in Teaching”. Even if you don’t literature related to teaching you could find some related to other professions/careers.

Response: Refer to response to (a). The new subsection added recent literature to the topic

4. Methods and Materials

1. So, what was the sampling technique? Random sampling or a mere point of saturation was employed?

Response: In our methods section, we stated emphatically that the we used census method (which involves a consideration of all possible research subjects in an investigation). Hence, it is difficult to understanding why the reviewer expects us to state whether we did simple random sampling or not. We stand to be corrected anyway, but we are certain that census method is another procedure used in getting research subjects for an investigation. But considering all subjects does not necessary all of the population would be involved. As could be observed in our study, the decision was to include all members of the population, by only 180 of them respondent to the questionnaire. It was a mere point of saturation either. The sampling technique is census method (census sampling). This is one of the many studies that employed this sampling technique:

BISWAS, S., & SEHGAL, V. K. (1987). Generalized probability model for estimating the proportion of the female population at different levels of fertility based on census sampling from a mixed population. International Journal of Systems Science, 18(10), 1909–1917. https://doi.org/10.1080/00207728708967163 .

Janatolmakan, M., Andaieshgar, B., Aryan, A., Jafari, F., & Khatony, A. (2019). Comparison of Depression Rate Between the First- and Final-Year Nursing Students in Kermanshah, Iran. Psychology Research and Behavior Management, 12, 1147–1153. https://doi.org/10.2147/PRBM.S238873

Raige, S. R., Stellefson, M., Chaney, B. H., & Alber, J. M. (2015). Pinterest as a Resource for Health Information on Chronic Obstructive Pulmonary Disease (COPD): A Social Media Content Analysis. American Journal of Health Education, 46(4), 241–251. https://doi.org/10.1080/19325037.2015.1044586

5. Results

1. The discussion could be strengthened by highlighting some plausible explanation why, for instance, subjective norms about teaching influences entry into the teaching profession. I urge authors to do the same for the remaining two factors of TPB (i.e., behavioural control— and behavioural beliefs)

Response: We would like to urge the reviewer to critically have a second look at the discussion section. Our discussion touches on why the three predictors had a significant association with Intention as well as the possible as the possible implications. Probably, the reviewer does not appreciate the style in which the arguments were presented. We have highlighted those areas in red.

6. Implications

1. Authors could have said something about the extent to which your findings can be generalized and highlighted contexts in which they could be applicable.

Response: We have addressed this issue. Under the managerial implication we have identified specific context the findings could be applied. We have stated that: “Though the study was conducted in a specific context of language teaching, which does not fully represent the broader context of teachers in other academic fields, the findings of the current study have implications for teachers in general. It provides insight into the career intention of prospective teachers in other fields, given that teachers' working conditions in Ghana, irrespective of the field of the teacher, are perceived to be suboptimal. The findings also imply any context where the profession suffers from a poor reputation as well as in contexts like Ghana African countries alike where the government seeks to improve teacher recruitment and retention.”

---

## [Decision Letter · Decision Letter 3]

23 Mar 2025

PONE-D-24-09444R3Who wants to be a teacher in Ghana?’ a structural equation modelling analysis of prospective language teachers’ behavioural intentions to pursue a career in teachingPLOS ONE

Dear Dr. Nyamekye,

Thank you for submitting your manuscript to PLOS ONE. After careful consideration, we feel that it has merit but does not fully meet PLOS ONE’s publication criteria as it currently stands. Therefore, we invite you to submit a revised version of the manuscript that addresses the points raised during the review process. A final round of proofreading is recommended to resolve residual grammatical/typographical errors. The reviewer requests a clearer discussion of the sampling technique (e.g., type of sampling, rationale, and limitations). Ensure this section is explicitly highlighted in the revised manuscript.

We look forward to receiving your revised manuscript.

Kind regards,

Dokun Iwalewa OIuwajana

Academic Editor

PLOS ONE

Journal Requirements:

Reviewers' comments:

Reviewer's Responses to Questions

**Comments to the Author**

1. If the authors have adequately addressed your comments raised in a previous round of review and you feel that this manuscript is now acceptable for publication, you may indicate that here to bypass the “Comments to the Author” section, enter your conflict of interest statement in the “Confidential to Editor” section, and submit your "Accept" recommendation.

Reviewer #2: (No Response)

2. Is the manuscript technically sound, and do the data support the conclusions?

Reviewer #2: Yes

3. Has the statistical analysis been performed appropriately and rigorously? 

Reviewer #2: Yes

4. Have the authors made all data underlying the findings in their manuscript fully available?

Reviewer #2: No

5. Is the manuscript presented in an intelligible fashion and written in standard English?

Reviewer #2: Yes

6. Review Comments to the Author

Reviewer #2: PONE

Reviewer’s Comments (Revision 3)

Title: ‘Who Wants to be a Teacher in Ghana?’ A Structural Equation Modelling Analysis of Prospective Teachers’ Behavioural Intention to Pursue a Career in Teaching

I have only these concerns. Best wishes to authors.

General Comments

1. Some proofreading is still required to fix a few errors.

Methods and Materials

1. Please kindly discuss the sampling technique used for the survey. Colour your discussion on the sampling technique to allow me to easily identify it. Thank you.

7. PLOS authors have the option to publish the peer review history of their article (what does this mean? ). If published, this will include your full peer review and any attached files.

**Do you want your identity to be public for this peer review?** For information about this choice, including consent withdrawal, please see our Privacy Policy .

Reviewer #2: **Yes: ** Solomon Aboagye

---

## [Author Response · Author response to Decision Letter 4]

25 Mar 2025

Response to Reviewers

Have the authors made all data underlying the findings in their manuscript fully available?

Reviewer #2: No

Response: During the submission, we provided a link to the data. The editorial board can provide this link. This is the link to the data https://data.mendeley.com/datasets/4j8y23wv2b/1 . This link is available in the submission system

Reviewer #2: PONE

Reviewer’s Comments (Revision 3)

Title: ‘Who Wants to be a Teacher in Ghana?’ A Structural Equation Modelling Analysis of Prospective Teachers’ Behavioural Intention to Pursue a Career in Teaching

General Comments

1. Some proofreading is still required to fix a few errors.

Response

The manuscript has been proofread. All grammatical issues have been fixed

Methods and Materials

1. Please kindly discuss the sampling technique used for the survey. Colour your discussion on the sampling technique to allow me to easily identify it. Thank you.

Response: We have offered a detailed discussion of the sampling method. We have reemphasized that the census method was used. We have provided justification for the use of census rather than sampling out of the entire population. (see page 11, last paragraph)

---

## [Editor Report · Decision Letter 4]

27 Mar 2025

PONE-D-24-09444R4Who wants to be a teacher in Ghana?’ a structural equation modelling analysis of prospective language teachers’ behavioural intentions to pursue a career in teachingPLOS ONE

Dear Dr. Nyamekye,

Thank you for submitting your manuscript to PLOS ONE. After careful consideration, we feel that it has merit but does not fully meet PLOS ONE’s publication criteria as it currently stands. Therefore, we invite you to submit a revised version of the manuscript that addresses the points raised during the review process.

Dear Dr. Ernest Nyamekye, i<small>t seems the revised document is not displaying properly in its current format. For clarity, kindly review the attached PDF, which reflects the correct version</small>

We look forward to receiving your revised manuscript.

Kind regards,

Dokun Iwalewa OIuwajana

Academic Editor

PLOS ONE
---

## [Author Response · Author response to Decision Letter 5]

28 Mar 2025

Response to Reviewers

Have the authors made all data underlying the findings in their manuscript fully available?

Reviewer #2: No

Response: During the submission, we provided a link to the data. The editorial board can provide this link. This is the link to the data https://data.mendeley.com/datasets/4j8y23wv2b/1 . This link is available in the submission system

Reviewer #2: PONE

Reviewer’s Comments (Revision 3)

Title: ‘Who Wants to be a Teacher in Ghana?’ A Structural Equation Modelling Analysis of Prospective Teachers’ Behavioural Intention to Pursue a Career in Teaching

General Comments

1. Some proofreading is still required to fix a few errors.

Response

The manuscript has been proofread. All grammatical issues have been fixed

Methods and Materials

1. Please kindly discuss the sampling technique used for the survey. Colour your discussion on the sampling technique to allow me to easily identify it. Thank you.

Response: We have offered a detailed discussion of the sampling method. We have reemphasized that the census method was used. We have provided justification for the use of census rather than sampling out of the entire population. (see page 11, last paragraph)

---

## [Editor Report · Decision Letter 5]

2 Apr 2025

Who wants to be a teacher in Ghana?’ a structural equation modelling analysis of prospective language teachers’ behavioural intentions to pursue a career in teaching

PONE-D-24-09444R5

Dear Dr. Ernest Nyamekye,

We’re pleased to inform you that your manuscript has been judged scientifically suitable for publication and will be formally accepted for publication once it meets all outstanding technical requirements.

Kind regards,

Dokun Iwalewa OIuwajana

Academic Editor

PLOS ONE
---

## [Editor Report · Acceptance letter]

PONE-D-24-09444R5

PLOS ONE

Dear Dr. Nyamekye,

I'm pleased to inform you that your manuscript has been deemed suitable for publication in PLOS ONE. Congratulations! Your manuscript is now being handed over to our production team.

Kind regards,

on behalf of

Asst Dokun Iwalewa OIuwajana

Academic Editor

PLOS ONE